# Weigh-in-Motion System Based on an Improved Kalman and LSTM-Attention Algorithm

**DOI:** 10.3390/s23010250

**Published:** 2022-12-26

**Authors:** Baidi Shi, Yongfeng Jiang, Yefeng Bao, Bingyan Chen, Ke Yang, Xianming Chen

**Affiliations:** 1College of Mechanical & Electrical Engineering, Hohai University, Changzhou 213022, China; 2Jiangsu Province Wind Power Structural Research Center, Nanjing 210003, China

**Keywords:** weigh-in-motion, deep learning, Kalman filter, time-series analysis

## Abstract

A weigh-in-motion (WIM) system continuously and automatically detects an object’s weight during transmission. The WIM system is used widely in logistics and industry due to increasing labor and time costs. However, the accuracy and stability of WIM system measurements could be affected by shock and vibration under high speed and heavy load. A novel six degrees-of-freedom (DOF), mass–spring damping-based Kalman filter with time scale (KFTS) algorithm was proposed to filter noise due to the multiple-input noise and its frequency that is highly coupled with the basic sensor signal. Additionally, an attention-based long short-term memory (LSTM) model was built to predict the object’s mass by using multiple time-series sensor signals. The results showed that the model has superior performance compared to support vector machine (SVM), fully connected network (FCN) and extreme gradient boosting (XGBoost) models. Experiments showed this improved deep learning model can provide remarkable accuracy under different loads, speed and working situations, which can be applied to the high-precision logistics industry.

## 1. Introduction

Weigh-in-motion (WIM) balance is of great interest in logistical sorting, which detects the cargo weight during the transport link, with an increasingly intensified contradiction between logistics demand and labor gap [1,2,3]. In addition, WIM technology is also widely used in vehicle weighing apparatus, dynamic railway scales, and automatic agricultural weight check with full advances of automation [4,5,6]. In recent decades, the development of WIM with a weigh data process has been one of the most important topics that have attracted the attention of various industries [7,8].

In general, the WIM system can be described in the following steps: (i) data acquisition by using pressure sensors and signal filter [9,10] and (ii) output a filtered signal by using linear regression (LR), an autoregression model [11,12], a machine learning (ML) algorithm [13,14] or a deep learning (DL) algorithm [15,16].

In the first aspect, sensors are usually located under a measurement table to measure the pressure in motion, and the data are sampled by pressure sensors. The signal is mainly sampled by pressure signal; however, its accuracy is affected due to the inevitable noise from motor vibration, measurement error, environmental factors [17], etc. Several achievements have been obtained for filtering noise in motion. A fuzzy logic estimator was used to filter noise during dynamic weighing by allocating a suitable weight to each sampling signal [18]. This method shows good performance during low speed. However, as the speed increases, the noise from the motor increases nonlinearly; thus, fuzzy logic LR models show poor performance under high speed and heavy load conditions. Autoregression (AR) models are used in WIM systems to mitigate this effect. Compared with the LR model, filtered data are decided by the current time and former signal; the filter can extract the feature from the time series. In addition, a system identification method is widely used in WIM systems [19] by using building a spring damping system. The filtered signal is partially decided by estimating the system identification equation and codetermined via the probability distributions of environmental noise. A Kalman filter (KF), as the typical system identification filter, is widely used in the WIM area [20,21,22]. The filtered signal is codetermined using a system-state matrix and sensor-sampling value at each time. By using a covariance matrix for dynamic state update, the KF has been proven to be a strictly linear optimal filter and is widely used in linear modeling.

The second problem concerns outputting weight by using the filtered signal. The mean value of the sampled signal is generally directly taken as output weight in low-precision industrial fields. However, it is inappropriate to assign the same weigh for each sampling point for the nonstable process of WIM; the lateral and longitudinal acceleration are uneven. The polynomial and exponential models minimize the fitting error by employing the least squares method [23]. It shows better performance than the traditional LR or the averaging model without adding too much operational complexity and is widely used in low-cost embedded devices. Nevertheless, these models are ineffective for high-speed and accurate WIM fields; the sampling time of the pressure signal is less than the system’s steady-state time, which makes the signal unable to reflect the transient response of the object’s mass precisely due to the nonstatic contact between the measurement tableboard and object [24]. In addition, the sampled pressure signal exhibits nonlinearity as the increasing of transmission speed and measurement weights. Each pressure sensor’s sampling frequency is usually larger than 1024 (Hz) during WIM processing. It is difficult to handle bulk and nonlinear data by using a traditional linear model. The model must have characteristics of nonlinearity and time-series processing capability to handle these problems. Currently, a deep learning algorithm centered on the convolution neural network (CNN), FCN and RNN has better performance than machine learning and statistical models in Big Data analysis, computer vision (CV), natural language processing (NLP) and many other fields [25]. FCN [26] and SVM [27] were used to perform data processing for WIM systems. Additionally, a few forefront achievements of deep learning have been reported. The sampled sequences strictly obey causal conditions [28] during the WIM, i.e., the present state is decided by the former situation and only affects subsequent states. LSTM [29] differs from the common regression models (SVM, XGBoost [30] and FCN); as a nonlinear autoregression structure, the causal features of the signal are considered. The measured data in the sampling interval are taken as input and a serial recursive structure is applied, which can tap the time-domain characteristics of the signal without destroying the temporal continuity of the signal sequence. Owing to these properties, RNN models are widely used in stock prediction, traffic-flow forecasting and time-series tasks. 

Attention mechanism is a remarkable achievement in deep learning. The performance of RNN can be greatly improved due to the simulation of human attention distribution. In which, the appropriate weight is allocated to each time during extensive data analysis [31,32,33].

Six DOF dynamic discrete-response models are built and the acceleration response under different load and belt velocity is analyzed in this paper. The corresponding improved KFTS is built by using the dynamic response system and actual sensor signals as the state estimation and measurement matrix. The key-value, attention-based LSTM data processing model is built and finds that the NAdam optimizer is optimal above SGD, RMSprop and Adamax. The measurement error of the SVM, XGBoost, FCN and attention-based LSTM models is compared. Drawbacks and conclusions are summarized at the end.

## 2. Establishment of the WIM Filter

### 2.1. Establishment and Analysis of the Dynamic Model

It is difficult to directly measure the accelerated response of pressure sensors due to the transfer table and kinds of limit protection devices. Normally, analytic models approximate the sensor response as equal to the instantaneous stress suffered by the tableboard. A typical WIM balance is shown in Figure 1.

In Figure 1, the main components of WIM scale are shown.

The response of the sensors and the motor’s cyclical electric force are used as input. The WIM system as its equivalent model with six degrees of freedom is shown in Figure 2. The system is poweredby one motor, and the measuring module comprises four pressure sensors to detect the instantaneous change in force on the table’s vertical direction. Consider the table’s steady center of gravity as the origin; the system’s generalized coordinate ***X*** can be expressed as follows:(1)$$X=\left\{x,x_{m},x_{1},x_{2},x_{3},x_{4}\right\}$$
where $x,x_{m},x_{1},x_{2},x_{3},x_{4}$ are the vertical displacements of table, motor, sensor-1, sensor-2, sensor-3 and sensor-4. Similarly, the velocity matrix and acceleration matrix by can be achieved by discrete differential operation.

In Figure 2, the structures of fixation and connection can be simplified as a spring damping system (*c*_1_, *c*_2_, *c*_m_, *c*_g_*, k*_1_, *k*_2_, *k*_g_*, k*_m_), and each objection’s stiffness and damping coefficient have been measured during the design process.

The vertical pressure is not directly affected to avoid damaging the pressure sensor (1~4). The inhibiting devices are installed between the table and sensor, which can be assumed as a spring damping system with a high damping coefficient and low stiffness; *m_t_* is the weight of measurement object. The acceleration of the table in the vertical direction can be measured using an acceleration sensor (sampling frequency is the same as the pressure sensor). The corresponding varying response *F_m_*(*t*) can be calculated using Newton’s second law.

The input force *Fm*(*t*) is driven by the motor. Its transient response directly determines the whole system’s steady state response as the only power source. The theory electric force of the driving motor in the vertical direction can be defined as follows:(2)$$F_{m}\left(t\right)=\{\begin{array}{l} \frac{T_{e}}{R}cos\left(\frac{a\pi }{30}t\right)-\frac{i^{2}\left(L_{min}+\frac{a\pi K}{30}\left(t-nT\right)\right)}{\sqrt{b^{2}{\left(\frac{\pi ar}{30}\right)}^{2}{\left(\frac{7T}{20}-\left(t-nT\right)\right)}^{2}}}\sin\left(\frac{a\pi }{30}t\right),nt\leq t\leq \left(n+\frac{7}{20}\right)T\\ 0,\;\;\;\;\;\;\;\;\;\;\;\;\;\;\;\;\;\;\;\;\;\;\;\;\;\;\;\;\;\;\;\;\;\;\;\;\;\;\;\;\;\;\left(n+\frac{7}{20}\right)Tt\leq t\leq \left(n+1\right)T,\left(n=0,1,2,\ldots \right) \end{array}$$
where *R*, *a*, *N_r_*, *b*, *I*, *K*, *T_e_* and *L_min_* are inner radius of stator, motor speed, rotor speed, minimum air gap length, winding current, rate of change in winding inductance with respect to position angle, rated torque and winding minimum inductance, respectively.

For the table’s shock and vibration in up and down directions, Equation (3) can be achieved based on the model shown in Figure 2.
(3)$$M\ddot{x}=F_{0}\left(t\right)-k_{m}\;\left(x-x_{m}\right)-2\;(k_{1}\left(x-x_{1}\right)-k_{2}\;\left(x-x_{2}\right)-c{\;}_{1}\left(\dot{x}-{\dot{x}}_{1}\right)-k_{2}\left(\dot{x}-{\dot{x}}_{2}\right))$$

The motor’s shock and vibration in up and down directions can be defined by Equation (4):(4)$$m_{o}{\ddot{x}}_{m}=F_{m}\left(t\right)-k_{m}\;\left(x_{m}-x\right)-c_{m}\;\left({\dot{x}}_{m}-\dot{x}\right)$$

The shock and vibration to pressure sensor-1 and pressure sensor-2 in up and down directions can be defined by Equation (5) if the differences in stiffness and damping caused by the assembly are ignored.
(5)$$m_{1}{\ddot{x}}_{m}=-k_{1}\;\left(x_{1}-x\right)-c_{1}\;\left({\dot{x}}_{1}-\dot{x}\right)-k_{g}\;\left(x_{1\;}-x\right)-c_{g}\;\left({\dot{x}}_{1}-\dot{x}\right)$$

A similar conclusion can be achieved in Equation (6) for sensor-3 and sensor-4:(6)$$m_{2}{\ddot{x}}_{2}=-k_{2}\;\left(x_{2}-x\right)-c_{1}\;\left({\dot{x}}_{2}-\dot{x}\right)-\;k_{g}\;\left(x_{2}-x\right)-c_{g}\;\left({\dot{x}}_{2}-\dot{x}\right)$$

The speed of the belt *v* (m/min) and weight *m* (kg) determine the steady-state response of the balance under actual engineering use for the object to be measured. The precise measurement is greatly affected due to object’s vibration during its transport. When the object makes contact, the balance table causes self-excited vibration under certain combinations of object weight and speed. This noise, which is a typical high frequency interference noise, greatly disturbs the pressure signal acquisition if it cannot be filtered.

Equations (3)–(6) can be expressed as a vector matrix:(7)$$M\cdot \ddot{x}\left(t\right)+C\cdot \dot{x}\left(t\right)+K\cdot x\left(t\right)=L\cdot U\left(t\right)$$
where $\ddot{x},\;\dot{x},x$ are the displacement velocity, accelerated velocity and displacement vectors; ***M***, ***C*** and ***K*** are the matrix of mass, damping and stiffness; ***L*** is a constant vector to locate random excitation and ***U***(*t*) is the nonstationary excitation vector.

The dynamic response, Equation (7), can be written in the form of a state equation:(8)$$\dot{X}\left(t\right)=g\left(X,\theta ,f\right)=A\cdot X\left(t\right)+B\cdot F\left(t\right)$$
where ***θ*** is the matrix of structure that contains the information of the system’s rigidity and damper. ***A*** is a 2*n*-by-2*n* matrix, ***B*** is a vector with length 2*n*, ***X*** is a vector with length 2*n* which contains the vector of acceleration and displacement. These can be described by the following equation:(9)$$\{\begin{array}{c} A=\left[\begin{array}{cc} 0 & I\\ -M^{-1}K & -M^{-1}C \end{array}\right]\\ B=\left[\begin{array}{c} 0\\ M^{-1} \end{array}\right]\\ F\left(t\right)=L\cdot U\left(t\right)\\ X\left(t\right)=\left[\begin{array}{c} x\left(t\right)\\ \dot{x}\left(t\right) \end{array}\right] \end{array}$$

Equation (9) is the continuous state equation; using $e^{-At}$ to multiply both sides can achieve the following relationship:(10)$$e^{-At}\left(X-A\cdot X\right)=e^{-A\cdot t}B\cdot F\left(t\right)$$

Integrating *t* over the interval (*t*_0_,*t*) in Equation (10), and substituting the initial conditions of *t*_0_, then the continuous equation of state solution can be achieved:(11)$$X\left(t\right)=\Phi \left(t,t_{0}\right)\cdot X\left(t_{0}\right)+\int_{t_{0}}^{t}\Phi \left(t,\tau \right)\cdot B\cdot F\left(\tau \right)d\tau$$
where $\Phi \left(t,t_{0}\right)=e^{A\left(t-t_{0}\right)}$ state transition matrix, which is used as the state estimation matrix in KF; Equation (11) is the standard solution of a continuous state equation, and its essence is equivalent to the Duhamel integral of a dynamical system.

However, in the WIM’s six DOF system, the actual signal is described as a discrete matrix and measured in each discrete sampled time by the pressure and acceleration sensors. Equation (11) needs to be transformed into discrete state equations to obtain the measurement matrix and corresponding covariance matrix:(12)$$X\left(t_{k+1}\right)=\Phi \left(t_{k+1},t_{k}\right)\cdot X\left(t_{0}\right)+\int_{t_{k}}^{t_{k+1}}\Phi \left(t_{k+1},\tau \right)\cdot B\cdot F\left(\tau \right)d\tau$$
where this equation is a discrete equation and determined by the sensor sampling frequency (1024 Hz), defined in the time interval $\Delta t\;=\;t_{k+1}-t_{k}$. Equation (12) can be rewritten as:(13)$$X_{k+1}=\Phi_{k}X_{k}+\Gamma_{k}F_{k},\Gamma_{k}=B\int_{0}^{\Delta t}e^{A\tau }d\tau$$
where $X_{k}$ and $F_{k}$ represent the state matrix at time *t*; $\Phi_{k}$ represents the state transition matrix from *t* to *t* + 1 and if the Δt is small enough, then $\Phi_{k}$ can be calculated approximately by the following Taylor expansion:(14)$$\Phi_{k}=e^{A\Delta t}=I+A\Delta t+\frac{1}{2!}A^{2}\Delta t^{2}+\cdots +\frac{1}{k!}A^{k}\Delta t^{k}\approx I+A\Delta t$$
where *I* = ***A***^0^ is a 2*n*-by-2*n* unit matrix; Equation (14) is the discrete state expression.

The discrete differential equation was solved in the Python 3.8.0 environment using the SymPy and SciPy Toolkits. Set the time step Δ*t* as 1/(1024)s, ***F****_t_*(*t)* is the real response-accelerated velocity under different belt speed. Figure 3 shows the accelerated velocity response of four pressure sensors under different weight and belt speeds.

In Figure 3, simulation results show:(1)The valid sampling point is reduced with increasing belt speed. Compared with Figure 3a,b, the signal is mainly influenced by low-frequency noise in low speed. Vibration noise shows more obvious effects, especially under high load due to sensor-1 and sensor-2 being closed to the cargo input side.(2)The signal indicates a nonlinearity and nonstationary process with increasing belt speed. More seriously, the measuring process is less than the system steady-state time with the decreasing sampling point, as shown in Figure 3c–f.(3)The pressure sensor is typically oscillatory underdamped; it is crucial to reduce the various internal and external noise from various working conditions. Self-excited vibration is mainly influenced by the genetic frequency, as a high frequency noise, it differs from other signals and can be filtered by a low-pass filter.

The Fourier and Butterworth filters are widely used in WIM systems. These are useful and easy to implement under the certain speed when the cutoff bandwidth is appropriate. However, the responses show different features under different belt speeds and loads, which must be carefully handled to avoid filtering the basic signal. The WIM system based on KF is proposed in the following section.

### 2.2. Algorithm of KFTS

The filtering steps of the improved KFTS are similar with traditional Kalman: (1) Discrete state equation ${\hat{X}}_{k}$ and transition matrix $\Phi_{k}$ are calculated by the WIM system’s estimation matrix. (2) Calculate the Kalman gain $K_{k}$ according to the actual measurement matrix $Z_{k}$, system process noise matrix ***W****_k_* ~***N*** (0,*Q_k_*) and measurement noise matrix $\varepsilon_{k}$~*N* (0,***R****_k_*). ***Q**_k_* and ***R****_k_* are the corresponding covariance matrices at time *t*, which are used to describe the environmental and random factor interference.

The filtering process of improved Kalman can be summarized in the following stages:

(1)Prediction: Calculate least-square (LS) state ${\hat{X}}_{k}$ based on the state transition matrix $\Phi_{k-1}$ and process noise matrix ***W****_k_*_−1_. The station of *k* + 1 can be calculated as follows:(15)$${\hat{X}}_{k+1}=\Phi_{k}\cdot {\hat{X}}_{k}+\Gamma_{k}\cdot F_{k}+W_{k}$$
where $\hat{X}\;\left(t_{k-1}\right)$ is the WIM system’s state estimation matrix at time *t_k_*_−1_; the state prediction covariance matrix can be described as:(16)$$\sum_{{\hat{X}}_{k}}=\Phi_{k}\cdot \sum_{{\hat{x}}_{k-1}}\cdot \Phi_{k}{}^{T}+Q_{k}/\alpha_{k-1}$$
where $\sum_{{\bar{X}}_{k}}$ is the WIM’s state-prediction covariance matrix at time *t_k_*;$\;\sum_{{\hat{x}}_{k-1}}$ is the WIM’s state estimation covariance matrix at time *t_k_*_−1_ and $\;\alpha_{k-1}$ is the adaptive factor at *k* − 1. (2)Measurement: Calculate the error vector ***i***(*t*_k_) based on the pressure sensor’s actual signal ***Z*** (*t*_k_), and ***i*** (*t*_k_) can be described as:(17)$$i\left(t_{k}\right)=Z\left(t_{k}\right)-J\left(t_{k}\right)\cdot \bar{X}\left(t_{k}\right)$$
where ***J*** is the Jacobian matrix of the measurement signal and can be calculated using a numerical differential.(3)Calculate Parameter: The theoretical innovation matrix ***C****_k_*, actual innovation matrix ${\hat{C}}_{k}$, adaptive factor $\alpha_{k}$ and Kalman gain ***K****_k_* can be calculated from the following equations:(18)$${\hat{C}}_{k}=\frac{1}{N}\sum_{i=1}^{N}i_{k-1}\cdot i_{k-1}^{T}$$
(19)$$C_{k}=E\left[i_{k-1}\cdot i_{k-1}^{T}\right]=J_{k}P_{k}J_{k}^{T}+R_{k}$$
(20)$$\alpha_{k}=\frac{c_{0}}{\left\|i_{k}\right\|}\cdot \left(\frac{\|{\hat{C}}_{k}\|/\sqrt{tr\left({\hat{C}}_{k}\right)}}{\|C_{k}\|/\sqrt{tr\left(C_{k}\right)}}\right)$$
where *N* is the length of time scale, determined by the sampled frequency, belt speed and length of the tableboard, i.e., the number of sampling points; $tr\;\;\left({\hat{C}}_{k}\right)$ is the trace of the innovation matrix. The updated value usually deviates from the actual value due to the noises that are from model and measurement error. It is necessary to apply the actual innovation matrix ${\hat{C}}_{k}$ to codetermine adaptive factor $\alpha_{k}$. When updating ***K****_k_*, the self-adapting equation is given as Equation (21):(21)$$K_{k}=\sum_{{\hat{X}}_{k}}\cdot J_{k}^{T}{\left(J_{k}\cdot P_{k}\cdot J_{k}^{T}+\alpha_{k}\cdot R_{k}\right)}^{-1}$$
***Q****_k_* and ***R****_k_* are reversely adjusted matrices to enhance the estimation accuracy. When $\alpha_{k}$ is a constant value, the KFTS degrades into the traditional extended Kalman filter.(4)Output: Calculate the filtered signal at time *k*:(22)$$X_{k+1}={\hat{X}}_{k+1}+K_{k}\cdot \left(Z\left(t_{k}\right)-J\left(t_{k}\right)\cdot \bar{X}\left(t_{k}\right)\right)$$

The traditional KF estimate of the state of the linear system is based optimally on the principle of recursive least variance estimation. Unlike the former, the extended Kalman filter extends this algorithm to nonlinear systems. The extended Kalman filter algorithm linearizes the model via the high-order Taylor expansion, which avoids the fitting error.

### 2.3. Performance Comparison

The system’s state matrix is defined by the system’s dynamic equation of mechanical structure (Section 2.1) and amended via the sampled signals in our improved KFTS algorithm. Figure 4 shows the performance under the different speeds (60, 90 and 120 m/min):

In Figure 4, our algorithm has been marked with the sign “*”. The fluctuation amplitude of the original gradually presents a nonlinear growth trend with the increase in transmission speed under the calibration weight of 20 kg. The acceleration is unavoidable during measurement due to the unevenness of the belt and balance, and the transmission error of the motor. The feature of the peak is used in the EMD filter, but the original state of the signal is excessively filtered and not applicable in the WIM system. The wavelet filter with a different base function shows poor performance under the high-speed state for the same reason. In addition, the effective measurement time(s) is directly affected by the belt speed (v), and the sampling time meets the following formula:(23)s=(l−2d)/v
where *l* is the length of balance and *d* is the length dimension in the speed direction.

Traditionally, the average value of filter data is taken as the output weight in general industrial area, the corresponding data handling process can be described as the following formula:(24)$$out=\sum_{i=1}^{s}\alpha_{i}m_{i}$$
where *out* is the final output weight, $\alpha_{i}$ is the weight of each sampling point (in the direct averaging method, the value is a constant 1/*s*).

The signal’s characteristics under different speeds show nonlinearity, according to Figure 4. Traditional linear algorithm is inappropriate to extract data feature and output exact weight. A deep learning-based model is built to handle the bulk signal and achieve an accurate weigh-in-motion result.

## 3. Building the Deep Learning Model

### 3.1. Training Dataset

Deep learning is a typical sample-learning model with feature self-organization. The completeness, adequacy and comprehensiveness of learning samples directly influence model performance. Factors of belt speed (v), load (m) and temperature C(T) are taken in our dataset to comprehensively contain various WIM’s working conditions. A three-factor and five-level (*L_35_*) orthogonal table was designed and is shown in Table 1.

Table 1 contains 25 combinations of the of *v*, *m* and *T* factors. Five hundred tests were conducted under each combination to avoid the randomness of a single measurement. Totally, 10,000 sample data were obtained. The entire experiment was conducted within the China Coal Research Institute to avoid environmental disturbances.

### 3.2. Residual Connection Module

The convolution operation at position *t* for a signal series ***x*** with length s is defined as follows:(25)$$y_{t}=\sum_{k=1}^{m}w_{k}x_{t-k+1}$$
where *m* is the length of convolution kernel and *w_k_* is the kernel’s value at position *k*. Additionally, the vector convolution can be defined as follows:(26)$$y={\left\{y_{i}\right\}}_{i=1}^{s-m+1}=w^{*}x$$

Several studies proved that the depth of the neural network (NN) directly determined the model’s feature extraction capability [25]. However, as the model layer is increased, the vanishing gradient problem becomes inevitable. The residual network [34] (ResNet) greatly improved the efficiency of information transmission via adding a short connection to the nonlinear convolution layer. The process is described by Equation (25):(27)h(x)=x+(h(x,θ)−x)
where the forward function is split into two parts: the identity function ***x*** and the residue function *h*(***x***) − ***x***. A nonlinear element composed of a neural network has sufficient ability to approximate the original objective function or residual function, but the latter is easier to learn in practice, according to the universal approximation theorem [35]. The data flow of the residual connection module is shown in Figure 5.

In Figure 5, θ is the learning parameter related to the convolution kernel, convolution channel and convolution times. The original characteristics of the signal are maintained in addition to creating constant functions to facilitate model training. Combining Equations (25)–(27), the forward function can be defined as follows:(28)$$F\left(x\right)=\sum_{i=1}^{sc}Relu\left(\sum_{j=1}^{Ni}w^{ij}*x+b^{ij}\right)+x\;$$
where $w^{ij}$ is the *j*-th filter’s weight in *i*-th layer $b^{ij}$ represents the bias, and they are both learnable parameters. *Ni* is the number of kernels in *i*-th layer.

### 3.3. Multiscale Feature Extraction

As described in Section 3.2, the data are processed by the residual connection module. The specific scale convolution kernel is used to extract specific features in the convolution network. Three kinds of kernel size are used to enrich the features of the signal, and the number of filters in each layer is 32, 96 and 32. The multiscale convolution layer combined with the residual connection module is shown in Figure 6.

In Figure 6, three convolution kernels with different scales are designed to extract data features. The corresponding features are fused by matrix stacking. It is inappropriate to assign the same weight to each feature during the entire WIM process. The strategy of weight distribution is crucial and directly influences the deep learning model’s performance.

### 3.4. Attention Mechanism Layer

For the linear model, the best weight distribution can be calculated by the least square method, maximum likelihood estimation and other probability estimation methods. The deep learning model is self-organizing and training is based on the error gradient; therefore, the traditional method is not applicable. The attention mechanism as a resource allocation strategy allocates limited computer resources on key features, which is the main method to improve model efficiency and solve information overload. A key-value pair-based attention layer was designed to allocate suitable weight to each channel and restructure the signal (the input and output have the same shape) to adapt the signal’s sequence feature. The channel-based attention module’s processing flow can be described as follows:

For the input channel sequence ***C***, the output sequence ***H*** has different shapes. The corresponding three sequences can be defined as a linear transformation of the input sequence by Equations (29)–(31):(29)Q=WQX∈ℝd3×N
(30)K=WKX∈ℝd3×N 
(31)V=WVX∈ℝd2×N 
where ***Q***, ***K*** and ***V*** denote the query vector sequence, the key vector sequence and the value vector sequence, respectively; *d*_3_ denotes the channel adjustment factor, and the nonlinear fitting capability of the model is enhanced when the value increases.

The output sequence can be calculated by the following equation:(32)$$h_{i}=att\left(\left(K,V\right),q_{i}\right)=\sum_{j=1}^{N}\alpha_{ij}v_{j}=\sum_{j=1}^{N}softmax\left(s\left(k_{j},q_{i}\right)\right)v_{j}$$

In addition, the weight vector α can be directly defined and dynamically adjusted according to the error gradient for each iteration via a Fully Connected Layer. By combining the information presented in Section 3.1, Section 3.2, Section 3.3 and Section 3.4, the signal feature handle path and the shape-change flow of the matrix can be described, as shown in Figure 7.

In Figure 7, the input signal was processed in two stages:

(1) Four convolution paths were designed in the multiscale convolution module to extract the input signal’s time-domain features except for the short-cut path. All paths are up-sampled twice to enrich the receptive field and down-sampled once to adjust the number of channels and deepen the layer depth. The short-cut path is aimed to build the identity mapping between input and the processed value, which is beneficial in the training stage and avoids the degeneration of the network. Additionally, channel max-pooling was used to reduce computational complexity and to smooth local noise.

(2) A key-value pair-based attention layer was designed to allocate suitable weight to each channel and restructure the signal (the input and output have the same shape).

### 3.5. Long Short-Term Memory (LSTM) Layers

In Section 3.1, Section 3.2, Section 3.3 and Section 3.4, the attention-based preprocessing filtering module was constructed, and in this section the data processing issues are discussed. This problem is equal to using filtered signals to output an object’s mass as accurately as possible. The back propagation (BP) algorithm, which can also be called as FCN, is widely used in weight-in-motion systems. However, as a supervised regression model, it can only build the nonlinear mapping connection with input and output, since time-series signals ***x*** with length *t* tend to exhibit time dependence and time-domain coupling. In this process, features are lost if autoregressive theory is not applied. If the look-back coefficient (LBC) is set to *k*, the autoregressive model can be described as:(33)$$y_{t}=c+\emptyset_{1}y_{t-1}+\emptyset_{2}y_{t-2}+\cdots +\emptyset_{k}y_{t-k}+e_{t}$$
where c is a constant term,$\;\emptyset_{1}\ldots \emptyset_{k}$ are the parameters of the model and $e_{t}$ is the vector of white noise.

Similarly, conclusions can be achieved in LSTM; each output *y_i_* (*k* ≤ *I* ≤ *t* − *k*) is influenced by former output *y_i_*_−*k*_ to y*_i_*_−1_ and input *x_i_* to extract the time sequence feature. These features were dropped in FCN, LR and common ML models limited to model structure. LSTM controls cell state using the forgetting gate, input gate and output gate. Sigmoid layer is used to dot the data to complete the addition and deletion of data features in these gate structures. It can effectively solve the problem of long-term dependence of RNN under bulk data: set the current time index as *t*; the data input is x; the predicted output is ***H***; and ***C*** is the memory unit. The LSTM structure is shown in Figure 8.

In Figure 8, *σ* is the sigmoid activation function; [***h****_t_*_−1_, ***x****_t_*] is the composite matrix of the network’s state parameters at *t* − 1 and the input ***x****_t_* at this round; ***W****_f_*, ***W****_t_*, ***W****_c_*, and ***W****_o_* are the weight matrices of the forgetting gate, input gate, output gate and state control gate. If the learnable bias is defined as ***b***, then these gates can be described by Equation (34):(34)$$\{\begin{array}{c} f_{t}=\sigma \left(W_{f}\left[h_{t-1},x_{t}\right]+b_{f}\right)\\ i_{t}=\sigma \left(W_{i}\left[h_{t-1},x_{t}\right]+b_{i}\right)\\ c_{t}=\sigma \left(W_{c}\left[h_{t-1},x_{t}\right]+b_{c}\right)\\ o_{t}=\sigma \left(W_{o}\left[h_{t-1},x_{t}\right]+b_{o}\right) \end{array}$$

By setting the weight matrix ***W*** for each gate in Equation (34) to 0 or 1, features of the input parameters can be effectively added and deleted through a dot product operation, and the output range of the sigmoid function is (0, 1). The final output of LSTM at time *t* is determined by the present input *x* and the output gate, which can be described as follows:(35)$$h_{t}=o_{t}tanh\left(c_{t}\right)$$

After the extraction of the time-series feature, a classic FCN is linked to the output weight. The entire model is shown in Figure 9.

In Figure 9, the training dataset is shown in Table 1; 80% of the dataset is used for model training and the remaining for model validation. The deep learning model consists of the feature extraction and aggregated output modules; the model’s training is described in the next section.

### 3.6. Model Training

The MultiConv1d-attention-LSTM model was built under Python 3.7.3 and the Pytorch 1.8.0. Toolkit. The training process is mainly about matrix calculation and gradient spread; the graphics processing unit (GPU)’s performance is remarkable faster than the center processing unit (CPU). By using CUDA11.0 and CUDNN 8.0.1, the model can be deployed in GPU. The computer configuration follows: CPU (i9-10900k), GPU (RTX3080) and RAM (32G 3200). Figure 6 shows the training loss and validation loss under different optimizations.

In Figure 10, diagrams show the training and validation loss curve with different optimizers under 100 iteration epochs. The optimizer determined the strategies of the learning rate. The SGD and SGD with Momentum update the learning parameters according to the loss gradient by randomly selected samples. This strategy is valid in the early period. However, the loss diverges in the end as the corresponding training process shown in Figure 10b,d. SGD optimizer amends the model according to the gradient. However, the loss showed the tendency of divergence due to the learning rate is fixed at each stage of the iteration. The models cannot attain an ideal state at the end of iteration. Adam is an optimizer that combines the advantages of Adadelta and RMSprop. Adam adaptively computes the learning rate for the training parameters via computation and stores each parameter’s exponentially decaying average of previous gradients and squared gradients. However, as shown in Figure 10a, the later period also shows the tendency of divergence. The loss of training and validation shown in Figure 10c indicates decreasing continuity when using NADAM. This method combined Adam and Nesterov accelerated gradient (NAG), which is superior to other optimizers; the final validation loss is convergent at 0.02.

## 4. Performance under a Practical Engineering Situation

### Building the Testing Environment

A TW155 dynamic scale in the factory was used to set-up a three-stage drive system to test each algorithm’s robustness and portability. The weights and the three-stage drive system (front and rear drive and measurement system) are shown in Figure 11. The weights of the measurement objects were measured using a high precision dynamic balance (static measurement error less than 0.001 g).

The measure errors for each model were measured under the data sets indicated above. Each target was measured 100 times under different speeds (30, 60, 90 and 120 m/min). The mean absolute error (MAE,$\;\bar{e}\left(\mathrm{kg}\right)$), mean relative error (MRE,$\;\bar{\delta_{e}}\;\left(\%\right)$) and mean maximum error (MME,$\;\bar{max}\;\left(\mathrm{kg}\right)$) were used to estimate each algorithm’s accuracy and stability. The performance of each algorithm is shown in Table 2, Table 3 and Table 4.

Each algorithm’s performance under different speeds is shown in Table 2, Table 3, Table 4 and Table 5. As the speed increases, each model’s MAE, MME and MRE shows nonlinear growth. The speed directly determined the valid sampling points and the stability of test objects in motion. The improved Kalman filter combined with the LSTM-attention algorithm shows the best performance under different speeds and loads.

## 5. Conclusions

An improved Kalman filter together with a dynamic equation of mechanical structure was used in this paper to estimate the system’s state matrix under different speeds. A deep learning base was built to process bulk data and output the weight. The results showed that:(1)The pressure signal’s noise indicates increasing nonlinearity, greatly affecting the accuracy and stability of the weight check in motion as the speed increases.(2)The improved Kalman filter can efficiently use the WIM system’s state matrix to estimate the system’s actual situation and filters the noise under different speeds.(3)Compared with the traditional models, a deep learning-based model decreases error and can greatly improve the system’s measurement accuracy.

Although the measurement accuracy of the WIM is improved with our Kalman and LSTM-attention algorithm, the personnel and resources necessary to construct the training sample set are costly. The calculated performance is also required in framework deployment and is not applicable for common embedded devices. Additional work needs to be carried out in the future:

(1) In logistics weighing, the sampling process can be approximately identified as a uniform velocity compared with vehicle scales, bridge vehicle weighing and other WIM fields. The applicability of the model to other WIM domains needs to be investigated.

(2) A method to simplify the neural network models is required so that the model can be deployed in low-cost embedded devices.

## Figures and Tables

**Figure 1 sensors-23-00250-f001:**
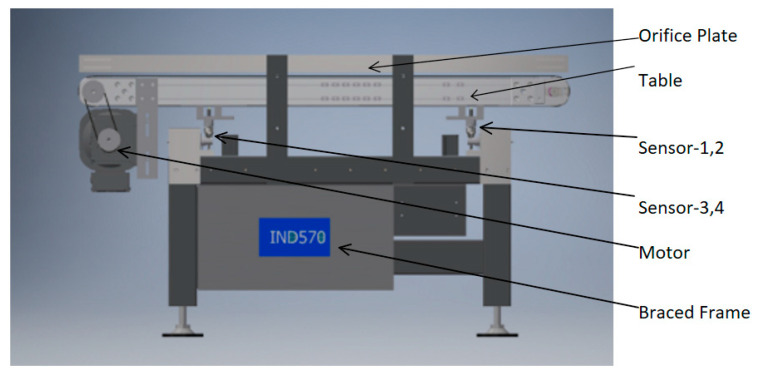
Structure of the WIM scale.

**Figure 2 sensors-23-00250-f002:**
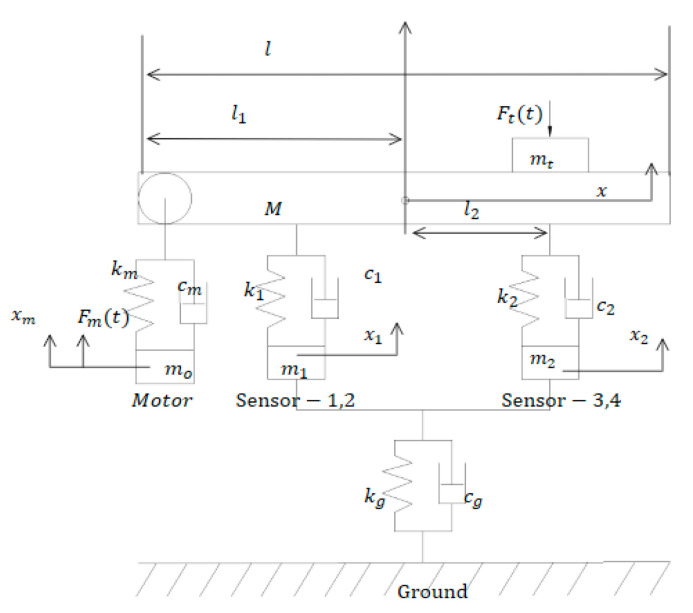
Equivalent model with six degrees of freedom.

**Figure 3 sensors-23-00250-f003:**
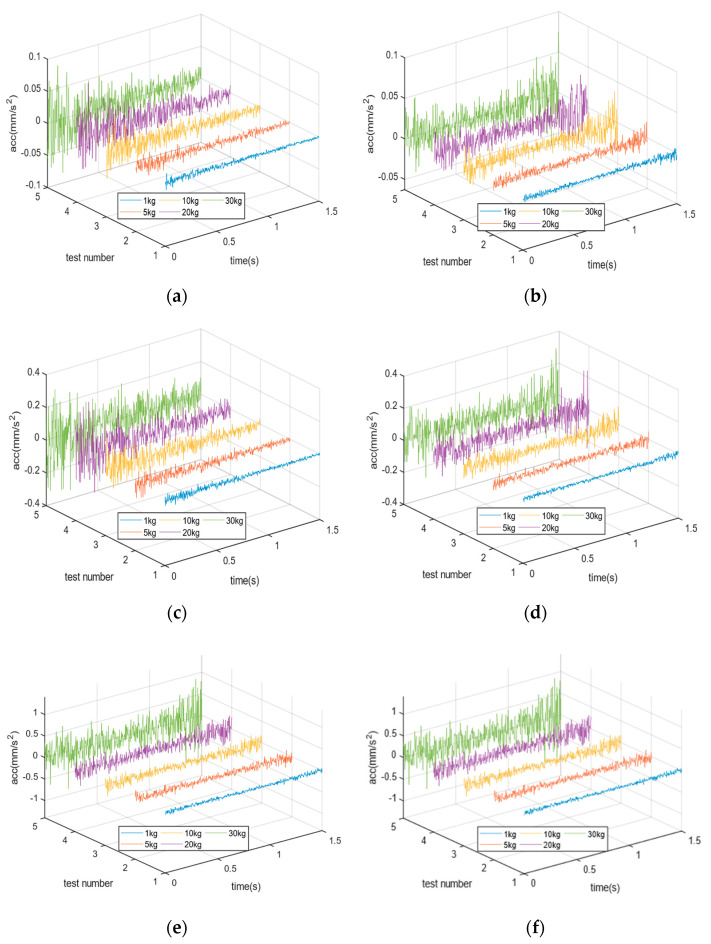
The time-varying accelerated velocity response of pressure sensors under different weight: (**a**) the response of sensor-1 and -2 under *v* = 45 (m/min), (**b**) the response of sensor-3 and -4 under *v* = 45 (m/min), (**c**) the response of sensor-1 and -2 under *v* = 90 (m/min), (**d**) the response of sensor-3 and -4 under *v* = 90 (m/min), (**e**) the response of sensor-1 and -2 under *v* = 120 (m/min), (**f**) the response of sensor-3 and -4 under *v* = 120 (m/min).

**Figure 4 sensors-23-00250-f004:**
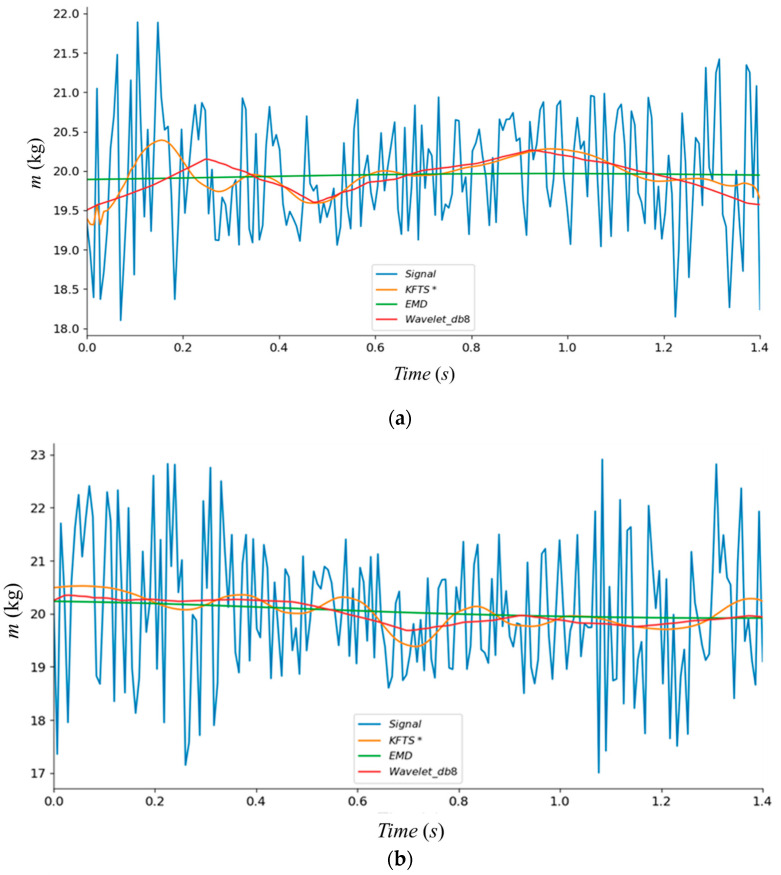
Comparison of filtering effects: (**a**) under the speed of 60 (m/min), (**b**) under the speed of 90 (m/min), (**c**) under the speed of 120 (m/min).

**Figure 5 sensors-23-00250-f005:**
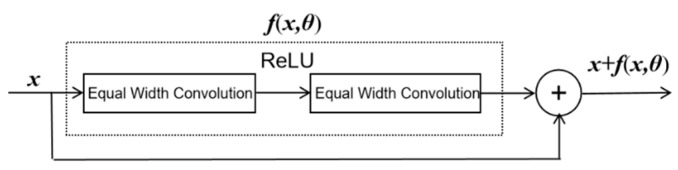
Residual connection module.

**Figure 6 sensors-23-00250-f006:**
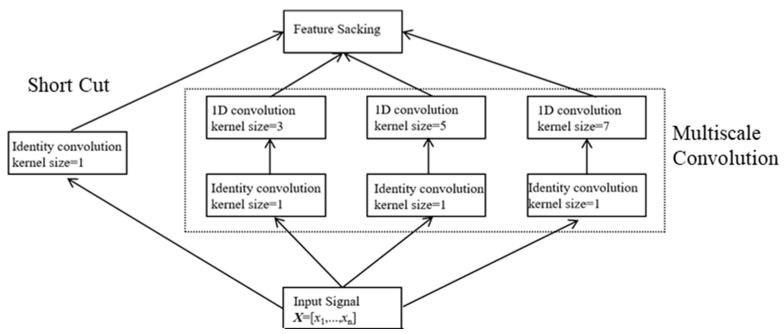
Multiscale convolution with residual connection.

**Figure 7 sensors-23-00250-f007:**
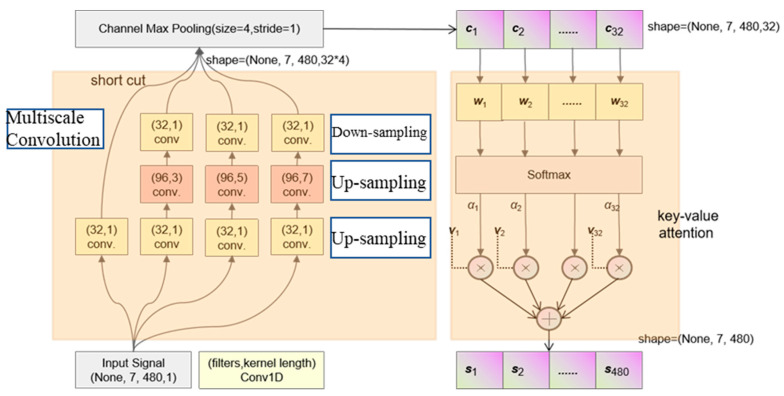
Flow of feature extraction.

**Figure 8 sensors-23-00250-f008:**
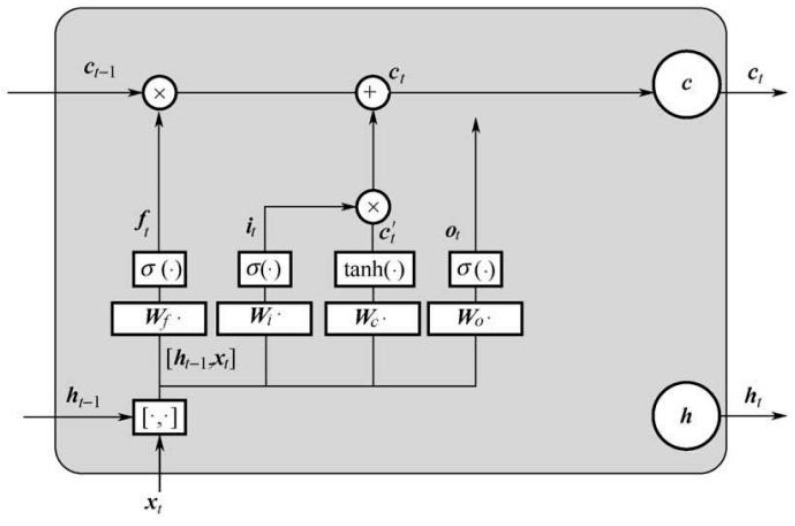
Structure of LSTM.

**Figure 9 sensors-23-00250-f009:**
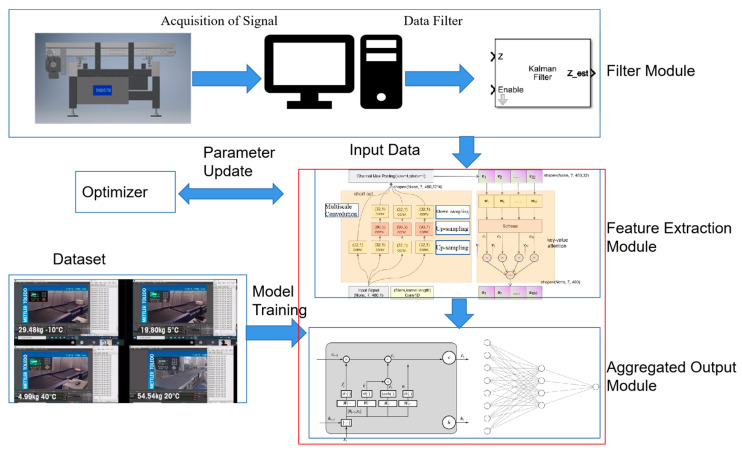
Structure of the entire model.

**Figure 10 sensors-23-00250-f010:**
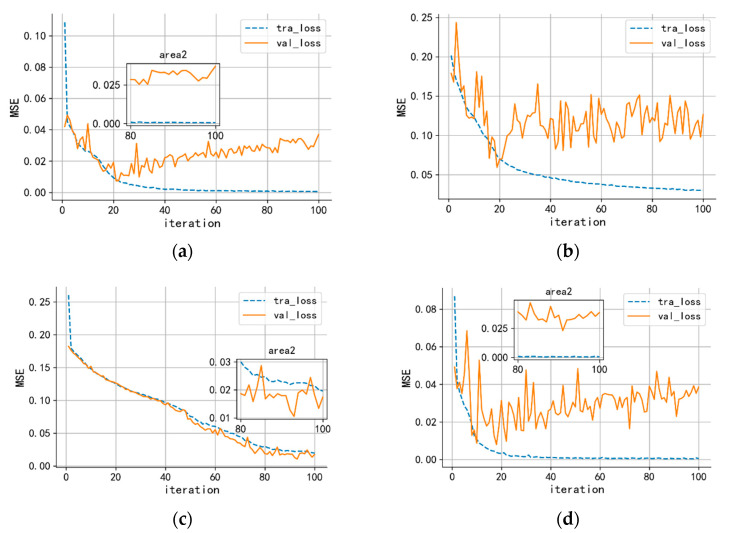
The loss of validation and training under different algorithm optimizers: (**a**) adaptive moment estimation (ADAM); (**b**) stochastic gradient descent (SGD); (**c**) Nesterov accelerated adaptive moment estimation (NADAM); (**d**) SGD with momentum.

**Figure 11 sensors-23-00250-f011:**
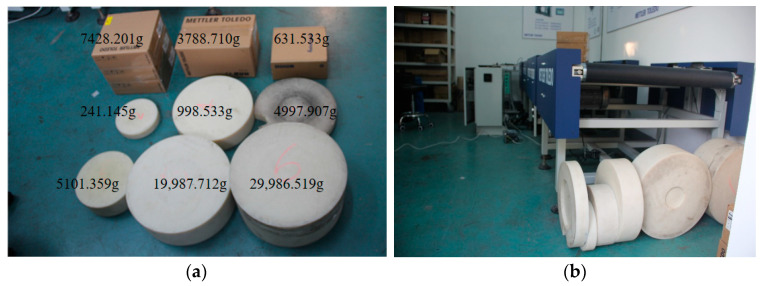
Testing environment: (**a**) measurement object and (**b**) three-stage drive system.

**Table 1 sensors-23-00250-t001:** The L_35_ orthogonal test table.

	A—Belt Speed (m/min)	B—Load Weight (kg)	C—Temperature (°C)
**A_1_B_1_C_1_**	**45**	**1**	**−10**
**A_1_B_2_C_2_**	**45**	**5**	**0**
**A_1_B_3_C_3_**	**45**	**10**	**10**
**A_1_B_4_C_4_**	**45**	**20**	**20**
**A_1_B_5_C_5_**	**45**	**30**	**40**
**A_2_B_1_C_2_**	**60**	**1**	**0**
**A_2_B_2_C_3_**	**60**	**5**	**10**
**A_2_B_3_C_4_**	**60**	**10**	**20**
**A_2_B_4_C_5_**	**60**	**20**	**40**
**A_2_B_5_C_1_**	**60**	**30**	**−10**
**A_3_B_1_C_3_**	**90**	**1**	**10**
**A_3_B_2_C_4_**	**90**	**5**	**20**
**A_3_B_3_C_5_**	**90**	**10**	**40**
**A_3_B_4_C_1_**	**90**	**20**	**−10**
**A_3_B_5_C_2_**	**90**	**30**	**0**
**A_4_B_1_C_4_**	**120**	**1**	**20**
**A_4_B_2_C_5_**	**120**	**5**	**40**
**A_4_B_3_C_1_**	**120**	**10**	**−10**
**A_4_B_4_C_2_**	**120**	**20**	**0**
**A_4_B_5_C_3_**	**120**	**30**	**10**

**Table 2 sensors-23-00250-t002:** SVM.

SVM	$\bar{e}\;\left(kg\right)$	$\bar{max\;}\left(kg\right)$	$\bar{\delta_{e}\;}\left(\%\right)$
v = 30 (m/min)	0.054	0.071	0.0700
v = 60 (m/min)	0.077	0.115	0.0997
v = 90 (m/min)	0.121	0.157	0.1568
v = 120 (m/min)	0.238	0.329	0.3084

**Table 3 sensors-23-00250-t003:** FCN.

FCN	$\bar{e}\;\left(kg\right)$	$\bar{max\;}\left(kg\right)$	$\bar{\delta_{e}\;}\left(\%\right)$
v = 30 (m/min)	0.074	0.091	0.0959
v = 60 (m/min)	0.107	0.183	0.1386
v = 90 (m/min)	0.143	0.294	0.1853
v = 120 (m/min)	0.278	0.410	0.3603

**Table 4 sensors-23-00250-t004:** XGBoost.

FCN	$\bar{e\;}\left(kg\right)$	$\bar{max}\;\left(kg\right)$	$\bar{\delta_{e}}\;\left(\%\right)$
v = 30 (m/min)	0.046	0.0053	0.0596
v = 60 (m/min)	0.097	0.0063	0.1256
v = 90 (m/min)	0.115	0.0074	0.1490
v = 120 (m/min)	0.218	0.0137	0.2824

**Table 5 sensors-23-00250-t005:** **Our Model**.

Our Model	$\bar{e}\;\left(kg\right)$	$\bar{max\;}\;\left(kg\right)$	$\bar{\delta_{e}}\;\;\left(\%\right)$
v = 30 (m/min)	0.034	0.041	0.0441
v = 60 (m/min)	0.057	0.091	0.0739
v = 90 (m/min)	0.084	0.132	0.1089
v = 120 (m/min)	0.108	0.194	0.1401

## Data Availability

Not applicable.

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
