# Peer review of "Weigh-in-Motion System Based on an Improved Kalman and LSTM-Attention Algorithm"

_sensors, 2022, doi:10.3390/s23010250_

Round 1
Reviewer 1 Report
In this manuscript, the authors investigate the weigh-in-motion system in logistics sorting based on improved Kalman and LSTM-attention algorithm. Overall the work falls within the scope of the journal and is interesting to readers. However, it needs improvement before it can be considered for publication.
It is necessary to clarify the limitation of the presented algorithm. For instance, in the bridge engineering, the WIM system needs to deal with much higher speed and much more weight (i.e., trucks) than that discussed in the manuscript. Will the algorithm be applicable then?
In Performance under Practical Engineering Situation, the training data sets and network structure should be detailed, while the results are reported directly in the manuscript without providing information about the model and data used.
Please, check the manuscript carefully as there are many writing errors, only a few are listed as follows:
WIN should be WIM in the abstract and the whole text.
Symbols in the tables are of very low resolution.
Line 428, grammar error.
Author Response
Thans for your advices. Corresponding Modification has been done according to your modifications. And some interesting experiments has been carried out. The following contents will answer your questions point-to-point.
Point 1: It is necessary to clarify the limitation of the presented algorithm. For instance, in the bridge engineering, the WIM system needs to deal with much higher speed and much more weight (i.e., trucks) than that discussed in the manuscript. Will the algorithm be applicable then?
Response 1: The limitation has been concluded in chapter 5. Some experimients have been conducted in dynamic forklift scale and is shown in Figure 1. The results did not meet expectations.
The results did not meet expectations, the preliminary analysis results are as follows:
- Although the test surface is relatively flat, the vibration from motor(Diesel Vehicles) and fork teeth is severe. It is hard to obtain the accuracy in motion, but check the missing of goods in motion can be realized.
- Driver's subjective factors, once the brakes are applied or the gas pedal is stepped on, the transient signal flutuations.
- The electric belt drive process is relatively smooth in WIM logistic scales compared with WIM forklift and other usage scenarios.
Point 2: In Performance under Practical Engineering Situation, the training data sets and network structure should be detailed, while the results are reported directly in the manuscript without providing information about the model and data used.
Response 2: The description of dataset is added in Chapter 3.1.And the used module of deep learning has been described as following:
1.Residual Connection Module is described in Chapter 3.2.
2.Multi-Scale Feature Extraction Module is described in Chapter 3.3
- Attention Mechanism Layer is described in Chapter 3.4.
- Long Short-Term Memory (LSTM) Layers is described in Chapter 3.5,
- The whole data handle processing is shown in Paper Figure 9.
- The model training process is individually written Chapter 3.6.
Details are shown in the following attachment.

Reviewer 2 Report
The authors have presented of the method for fast and accurate weight determination of an object during transmission. The presented idea is interesting and practically important. The authors propose to use the six degrees-of-freedom mass-spring-damp model with application the Kalman filter with-Time-Scale algorithm to noise filter out. A few deep learning algorithms are investigated to improve the measure accuracy and decrease the errors. However, the article is very poorly written, with many linguistic errors, making it difficult to follow the authors' main arguments. It was probably computer-translated, without any final proofreading, as evidenced by errors even in references list.
Author Response
Dear Reviewers,
The english in this paper has been improved during these day via my mentor.
Round 2
Reviewer 1 Report
The authors have made substantial changes to the manuscript.